# Regional nonsense constraint offers biological and clinical insights into genetic disease

Alexander J. M. Blakes [1] ✉, Nicola Whiffin[2,3,4], Colin A. Johnson [5], Jamie M. Ellingford [1,6,7,8] & Siddharth Banka [1,6,8] ✉

Reliably predicting the molecular impact of premature termination codons (PTCs) is essential for the clinical interpretation of "loss-of-function" variants in human disease. Measures of selective constraint can identify genes and genomic regions which are intolerant to deleterious genetic variation. However, existing loss-of-function constraint metrics do not comprehensively account for nonsense-mediated mRNA decay (NMD), a quality control pathway which critically regulates PTCs. Here, we use sequencing data from 730,947 individuals to develop an NMD-informed regional nonsense constraint metric. We find 2764 genes with significant regional nonsense constraint, including 641 known autosomal dominant disease genes. Using sequencing data in 32,260 trios from three rare disease cohorts, we find that de novo nonsense and frameshift variants are 9.5-fold enriched and associated with up to 5.9-fold higher odds of diagnosis in constrained regions versus unconstrained regions. We use these data to identify 22 candidate disease genes with clusters of de novo variants in constrained regions. These findings enhance clinical variant interpretation, deliver mechanistic insights in human disease, and empower the discovery of novel disease genes.

Predicted 'loss-of-function' (pLoF) variants span a variety of genomic variant classes, from single-nucleotide changes to large structural variants, which disrupt the expression or function of a gene product. An important class of pLoF variants is those that introduce premature termination codons (PTCs), including nonsense, frameshift, and splicing variants. These variants are critically regulated by nonsense-mediated decay (NMD), an mRNA quality control pathway that selectively degrades transcripts carrying a PTC[1].

Classically, NMD is triggered by transcripts which contain an exon junction complex (EJC) downstream of an in-frame stop codon. The EJC is a multi-protein complex deposited by the spliceosome ~24nt upstream of most exon–exon junctions, which is usually displaced by the ribosome during a pioneer round of translation[2]. However, EJCs more than 30nt downstream of a stop codon may remain bound to the transcript. Interactions between the EJC, a terminating ribosome, and the NMD factors UPF1 and SMG1, trigger NMD and ultimately the degradation of the variant mRNA[3]. EJC-independent NMD pathways have also been described[2].

PTCs predicted to trigger NMD are broadly expected to lead to LoF through targeted destruction of the variant transcript. However, some PTCs may escape NMD, such that the variant transcript can persist and be translated into a truncated protein. Besides LoF, these

[1]Manchester Centre for Genomic Medicine, Division of Evolution and Genomic Sciences, School of Biological Sciences, Faculty of Biology, Medicine and Health, University of Manchester, Manchester, UK. [2]Big Data Institute, University of Oxford, Oxford, UK. [3]Centre for Human Genetics, University of Oxford, Oxford, UK. [4]Program in Medical and Population Genetics, Broad Institute of MIT and Harvard, Cambridge, MA, USA. [5]Division of Molecular Medicine, Leeds Institute of Medical Research, University of Leeds, Leeds, UK. [6]Manchester Centre for Genomic Medicine, Manchester University NHS Foundation Trust, Health Innovation Manchester, Manchester, UK. [7]Genomics England Ltd, London, UK. [8]These authors contributed equally: Jamie M. Ellingford, Siddharth Banka. ✉e-mail: alexander.blakes@manchester.ac.uk; siddharth.banka@manchester.ac.uk

truncated proteins may give rise to gain-of-function (GoF) or dominant-negative effects[4,5]. Indeed, some truncations may have no measurable effect on protein function. NMD efficiency is largely determined by the position of the PTC within the mRNA[6]. PTCs in the final exon of a transcript have no downstream EJC and therefore do not trigger EJC-mediated NMD[7]. For PTCs in the most 3' 50-55nt of the penultimate exon, the final EJC may be displaced by the translating ribosome[7]. PTCs in the first ~150 nt of the CDS may allow downstream translation re-initiation, resulting in an N-truncated protein[8]. Finally, PTCs in very long exons (greater than approximately 400 nt), particularly at the 5' end of the exon, are less likely to trigger NMD, possibly due to weakened interactions between the terminating ribosome and the downstream EJC[9].

Accounting for NMD is critical to the interpretation of nonsense, frameshift, and splicing variants in human disease. Recent clinical variant interpretation guidance uses susceptibility to NMD as the key discriminator for variant classification: PTCs which trigger NMD are generally considered more deleterious than those which do not[10]. However, NMD can mediate important genotype–phenotype correlations in genetic disease[11], and NMD-escaping variants can be highly deleterious. For example, heterozygous truncating variants specifically in the last exon of *FOSL2* cause a syndrome of aplasia Cutis and enamel dysplasia[12] (MIM 620789), and heterozygous truncating variants in the final exon of *SOX10* cause a more severe presentation (PCWH syndrome, MIM 609136) compared to true LoF variants (Waardenburg Syndrome, MIM 613266) through a dominant negative mechanism[13].

The proliferation of genomic sequencing data from large population cohorts has empowered the study of negative selection in the human genome. Negative selection removes deleterious variants which reduce reproductive fitness from the population. Variant classes, genomic regions, or genes which are depleted of variation compared to expectation are said to be under selective constraint[14–20]. Measures of constraint against pLoF variants in particular are widely used in clinical variant interpretation[21] and novel disease discovery[22]. However, existing pLoF constraint metrics do not comprehensively account for NMD.

Here, we show that PTCs in over a third of the coding genome are predicted to escape NMD. We leverage biobank-scale sequencing data to quantify regional constraints against nonsense variants and show that the strength of selection against PTCs differs across NMD regions. We find 2764 genes with significant regional nonsense constraint, 12–19% of which are currently not annotated as constrained using existing metrics. We also find that de novo variants in constrained regions are highly enriched and associated with higher rates of clinical diagnosis in rare disease trios. Finally, we prioritise 22 candidate disease genes in which de novo variants cluster in constrained regions. These results offer mechanistic insights into genome biology, improve the clinical interpretation of pLoF variants, and empower the discovery of novel diseases and disease genes.

## Results

### Variants in over a third of the coding genome are predicted to escape NMD

To understand the impact of NMD-escaping PTCs on genetic disease (Fig. 1a) we first annotated predicted NMD-escape regions in the canonical transcripts of all human protein-coding genes. Because the position of the PTC in the mature mRNA is the major determinant of NMD efficiency[6], we used positional rules to define potential 'NMD-escape' regions[23] (Fig. 1b). These included 'start-proximal' regions[6,9,24] (<150nt from the canonical start codon), 'long exon' regions[6,9] (>400nt upstream of a splice donor site) and 'distal' regions[7] (the final exon, and the most 3' 50nt of the penultimate exon). Protein-coding sites outside of these regions were defined as 'NMD-target' regions. Overall, we found that 38.7% of coding sequence (CDS) positions in canonical transcripts occur in predicted NMD-escape regions (Fig. 1c). To gauge

the clinical relevance of these annotations, we examined the frequency and classification of nonsense and frameshift variants in ClinVar[25]. Compared to NMD-target regions, nonsense and frameshift variants in start-proximal and distal regions are both significantly under-reported (Fig. 1d, Chi-squared goodness-of-fit test, Bonferroni $P < 0.0083$) and significantly enriched for variants of uncertain significance (VUS) in ClinVar (Fig. 1e, two-sided proportions Z test, Bonferroni $P < 0.0083$). These data highlight that the clinical interpretation of nonsense and frameshift variants predicted to escape NMD is a substantial challenge.

### Allele-specific expression validates positional NMD rules in germline variants

To validate these NMD rules in the context of germline variation in rare disease probands, we performed an allele-specific expression (ASE) analysis in RNA-Seq data from blood in 5132 individuals in the 100,000 Genomes Project (100kGP). The alternate allele ratio was significantly greater for rare synonymous variants (negative control; mean variant allele fraction (VAF) = 0.487) than for rare nonsense variants in NMD-target regions (mean VAF 0.383, two-sided Mann Whitney $U$ test $P < 1 \times 10^{-16}$; Fig. 2). The alternate allele ratio was also significantly higher for rare nonsense variants across all NMD-escape regions versus NMD-target regions (start-proximal mean VAF = 0.461, $P < 1 \times 10^{-16}$; long exon=0.438, $P = 4.74 \times 10^{-7}$; distal = 0.486, $P < 1 \times 10^{-16}$). The mean alternate allele ratio for rare nonsense variants in distal regions approached that of rare synonymous variants (0.486 and 0.487, respectively). These data imply that, on aggregate and consistent with expectations, NMD efficiency is measurably weaker in our predicted NMD-escape regions than in predicted NMD-target regions.

### The strength of selection against PTCs differs across NMD regions

Next, we quantified the overall strength of selection against subgroups of nonsense variants using the mutability-adjusted proportion of singletons (MAPS)[14] in exome sequencing data from 730,947 individuals in gnomAD v4.1[16] (Fig. 3a). We found that nonsense variants in all three NMD-escape regions are constrained, but the constraint was significantly weaker for start-proximal (MAPS = 0.083) and distal regions (MAPS = 0.066) compared to NMD-target regions (MAPS = 0.112) (two-sided proportions Z test, Bonferroni $P < 0.0024$). This is consistent with previous observations that NMD-escaping variants are under weaker negative selection than NMD-targeted variants[6]. Conversely, nonsense variants in long exon regions are significantly more highly constrained (MAPS = 0.130, two-sided proportions Z test, Bonferroni $P < 0.0024$). These data show that the strength of selection against nonsense variants can markedly differ depending on the position of the variant in the transcript.

### 2764 transcripts show regional constraint against PTCs

To quantify regional constraint against nonsense variants within individual transcripts, we used mutation rate estimates from Roulette[26], scaled to the gnomAD v4.1 exome sequencing data, to model the expected number of single-nucleotide variants (SNVs) in a given transcript or NMD region. For each transcript, region, and single nucleotide variant consequence (synonymous, missense, or nonsense), we calculated the ratio of observed/expected variants (O/E) (Supplementary Fig. 1), and the upper 95% confidence interval for this value (OE95) (Fig. 3b). Transcript-level OE95 values correlate strongly with loss-of-function observed/expected upper fraction (LOEUF) scores[16] from gnomAD v4.1 (Spearman rho = 0.871, $P < 10^{-8}$) thus validating our approach (Supplementary Fig. 2).

We used these OE95 statistics to identify 2668/19,676 (13.6%) canonical transcripts with significant transcript-level nonsense constraint (OE95 < 0.6, FDR < 0.05, one-sided Poisson-binomial test). Applying these models to predicted NMD-target and NMD-escape regions, we found 2764/19,676 (14.0%) transcripts with significant

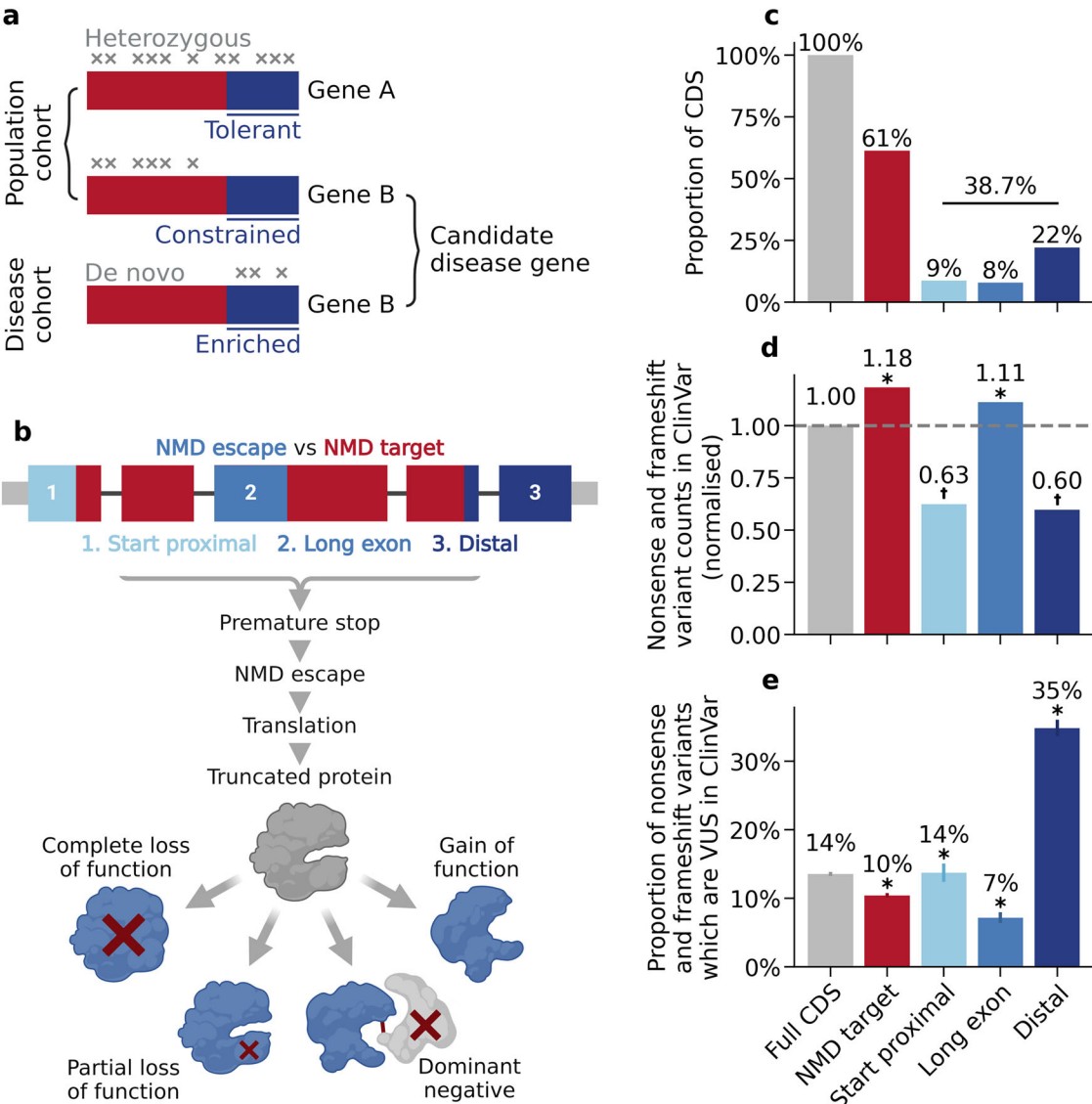

**Fig. 1 | PTCs in predicted NMD-escape regions are challenging to interpret clinically. a** Graphical abstract of this study. We identified NMD regions in protein-coding genes which are significantly constrained for nonsense variation in population sequencing cohorts. Within these constrained regions, we explored patterns of de novo protein-truncating variation in rare disease cohorts to prioritise novel candidate disease genes. (Created in BioRender. Blakes, A. (2024) BioRender.com/g72m993). **b** The molecular consequences of NMD escape. Transcript diagram (top) showing coding exons (thick boxes), with NMD-target regions (red) and NMD-escape regions (blue). Introns (black line) and UTRs (narrow grey boxes) are shown (not to scale). Transcripts carrying PTCs in certain regions may escape NMD. The transcript is translated into a truncated protein product. Protein truncation may have diverse consequences. (Created in BioRender. Blakes, A. (2024) BioRender.com/g72m993). **c** The genomic footprint of NMD regions in the canonical transcripts of 19,676 human protein coding genes (Methods). The proportion of the CDS occupied by predicted NMD-escape regions is shown. **d** Ascertainment of

nonsense and frameshift variants in ClinVar. All nonsense and frameshift variants in ClinVar are included, irrespective of ACMG classification (full CDS $N = 47,211$; NMD-target $N = 34,255$; start proximal $N = 2564$; long exon $N = 4147$, distal $N = 6245$). The absolute variant count in each region is normalised by the genomic footprint of that region. Asterisks indicate categories which are significantly different across all pairwise comparisons (Chi-squared goodness-of-fit test with Bonferroni correction for six tests at alpha=0.05, $P < 0.0083$). Dagger symbols indicate categories which are significant for all pairwise comparisons, except other categories with a dagger symbol. **e** The proportion of nonsense and frameshift variants classified as variants of uncertain significance (VUS) in ClinVar. Error bars show 95% confidence intervals. Asterisks indicate categories that are significantly different across all pairwise comparisons (two-sided proportions Z test with Bonferroni correction for six tests at alpha=0.05, $P < 0.0083$). Source data are provided as a Source Data file. Exact P values for **d** and **e** are provided in the Source Data file.

nonsense constraint in at least one region (Fig. 3c). These included 2216 (11.3%) transcripts constrained in NMD-target regions, 101 (0.51%) in start-proximal regions, 396 (2.0%) in long exon regions, and 612 (3.1%) in distal regions (FDR < 0.05, one-sided Poisson-binomial test) (Fig. 3c). 507 transcripts (2.6%) are constrained in more than one region (Supplementary Fig. 3).

We compared our results with LOEUF[16], a widely used transcript-level constraint metric. We found that 318/2764 (11.5%) transcripts with regional nonsense constraint are unconstrained at the transcript level

(pLI[14] =< 0.9 and LOEUF[16] >= 0.6, $N = 311$) or are missing constraint annotations ($N = 7$) in gnomAD v4.1 (Fig. 3c, Supplementary Fig. 4). We also compared our OE95 metric with GeneBayes[18], a recently developed novel constraint metric which uses an evolutionary model combined with machine learning on gene features. We find 529/2764 (19.1%) genes with significant regional nonsense constraint, but which are not prioritised by GeneBayes (Supplementary Fig. 5). Of these, 80 (15.1%) are already known autosomal dominant disease genes in OMIM[27] (Supplementary Fig. 6). These results show that our NMD-

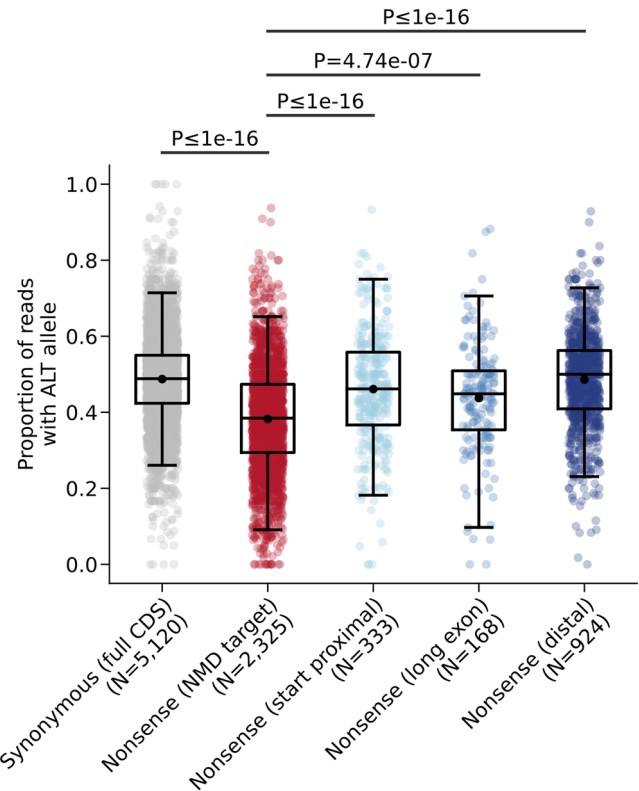

**Fig. 2 | Allele-specific expression of nonsense variants in NMD regions.** Alternate allele fractions for singleton variants in RNA-Seq data from blood in 7829 individuals in 100KGP. Synonymous variants were randomly down-sampled, without replacement, to match the total number of nonsense variants prior to the allele-specific expression analysis. Box and whiskers show the 0.025, 0.25, 0.5, 0.75, and 0.975 quantiles of the data. Solid black points show the means of the data. *P* values from two-sided Mann-Whitney *U* tests are shown. Source data are provided as a Source Data file.

informed constraint metric prioritises and distinguishes between transcripts with specific regional intolerance to PTCs.

## Constrained regions are highly conserved between species

We then compared the regional nonsense constraint with orthogonal measures of functional importance. Constrained regions were significantly more highly conserved across species (phyloP[28]) and had higher in silico predictions of functional importance (AlphaMissense[29]) and pathogenicity (Combined Annotation Dependent Depletion, CADD[30]) than unconstrained regions (Supplementary Fig. 7 two-sided Welch's T-tests, Bonferroni $P < 10^{-8}$). Surprisingly, relative expression (base-level pext[31]) was comparable between constrained and unconstrained regions (Supplementary Fig. 7d). Together, these results suggest that constrained regions are more likely to harbour functionally important sites.

Next, we explored the intersection of NMD regions with Pfam domains[32]. NMD-target regions are more likely to contain at least one Pfam domain (91.5%) than either start proximal (54.2%), long exon (49.8%), or distal regions (68.2%) (Supplementary Fig. 8). Constrained NMD-target regions and distal regions are significantly enriched for overlap with at least one Pfam domain (two-sided, two-sample Z test of proportions, $P < 1 \times 10^{-16}$ and $P = 1.67 \times 10^{-4}$, respectively). We observed no significant difference in the occurrence of Pfam domains in constrained or unconstrained start-proximal or long-exon regions. These results suggest that PTCs in constrained distal and NMD-target regions are more likely to disrupt a Pfam domain.

Discrepancies in within-species constraint and between-species conservation can highlight evolutionarily recent functional elements and

clade-specific biology. We therefore explored patterns of base-level conservation (phyloP) in nonsense constrained regions. We found that constrained long exon and distal regions have highly divergent conservation scores compared to constrained NMD-target regions (percentile difference 0.31 and 0.29 versus 0.17, respectively) (Supplementary Fig. 9), suggesting that these constrained elements tend to be evolutionarily more recent. Indeed, we find 21 genes with extreme discrepancies (constraint >90th centile, conservation <10th centile), of which nine are in distal regions (Supplementary Data 1). For example, the critical pluripotency factor *NANOG* has a highly constrained (OE95 0.43, 98th centile) and weakly conserved (median phyloP 0.46, 8th centile) distal region; the last exon encodes an intrinsically disordered region with strong transactivation activity[33,34]. These data highlight that regional constraints can prioritise gene regions that may be weakly conserved through evolution but are functionally important.

## Regional nonsense constraint highlights the clinical importance of PTC location

Non-uniform distributions of PTCs can inform disease mechanisms and clinical phenotypes[4,35,36]. We therefore examined patterns of nonsense and frameshift variation in transcripts with differential constraint across NMD regions (Fig. 4). We find genes with strong transcript-level nonsense constraint (pLI > 0.9, LOEUF < 0.6) but with specific NMD regions that are tolerant to PTCs. Examples include the unconstrained long exon region in *AJAP1* (Fig. 5a) and the unconstrained start proximal and long exon regions in *GATA6*, which has been previously described[35] (Supplementary Fig. 10). We also find genes with weak transcript-level nonsense constraint (pLI < =0.9, LOEUF > = 0.6) but strong nonsense constraint in NMD-target regions, such as *BTF3* (Fig. 5b) and *RSPO3* (Supplementary Fig. 11). A clinically relevant example is *FOXF1* (Alveolar capillary dysplasia[37], MIM 265380), which has a clear bimodal distribution of nonsense and frameshift variants in gnomAD, whereas pathogenic nonsense variants in ClinVar have an inverse distribution (Fig. 5c). Finally, we find genes with weak transcript-level nonsense constraint (pLI < =0.9, LOEUF > = 0.6) but strong nonsense constraint in NMD-escape regions, such as the start-proximal region of *CBX7* (Supplementary Fig. 12), the long exon region of *PROSER1* (Supplementary Fig. 13), and the distal region of *IRF2BP2* (Supplementary Fig. 14). Genes in the protocadherin alpha cluster (*PCDHA1* to *PCDHA13*) are a striking example; all have weak transcript-level constraint (LOEUF > 0.8) but strong nonsense constraint in distal NMD-escape regions (Fig. 5d, Supplementary Fig. 15). The final three exons of *PCDHA1* encode a common cytoplasmic domain shared by all genes in the cluster[38,39]. Distal truncations, therefore, appear highly deleterious, whereas the non-overlapping long first exons of each gene are tolerant to nonsense and frameshift variants. We observe a similar trend in the protocadherin gamma gene cluster (Supplementary Fig. 16). A clinical example of nonsense constraint specific to NMD-escape regions is *ASXL3* (Bainbridge-Ropers syndrome[40], MIM 615485) (Supplementary Fig. 17). In *ASXL3*, pathogenic nonsense and frameshift variants cluster strongly in the constrained long exon and distal regions[41] (nonsense OE95 = 0.17 and 0.21, respectively), however, the NMD-target region is relatively tolerant to PTCs (OE95 = 0.67). Collectively, these results show that regional nonsense constraint captures clinically and biologically relevant patterns of truncating variation. We expect these results to improve the clinical interpretation of PTCs in monogenic disease genes.

To explore the utility of regional nonsense constraint in the clinical setting, we examined diagnostic outcomes for carriers of de novo nonsense or frameshift variants in 13,908 rare disease trios from the 100,000 Genomes Project[42,43]. Individuals carrying a nonsense or frameshift variant in a constrained transcript had significantly greater odds of receiving a diagnosis (OR 4.95, 95% CI 4.19–5.85) than those without (Fig. 6a). Similarly, carriers of de novo nonsense or frameshift variants in constrained NMD-target, long exon, or distal regions have

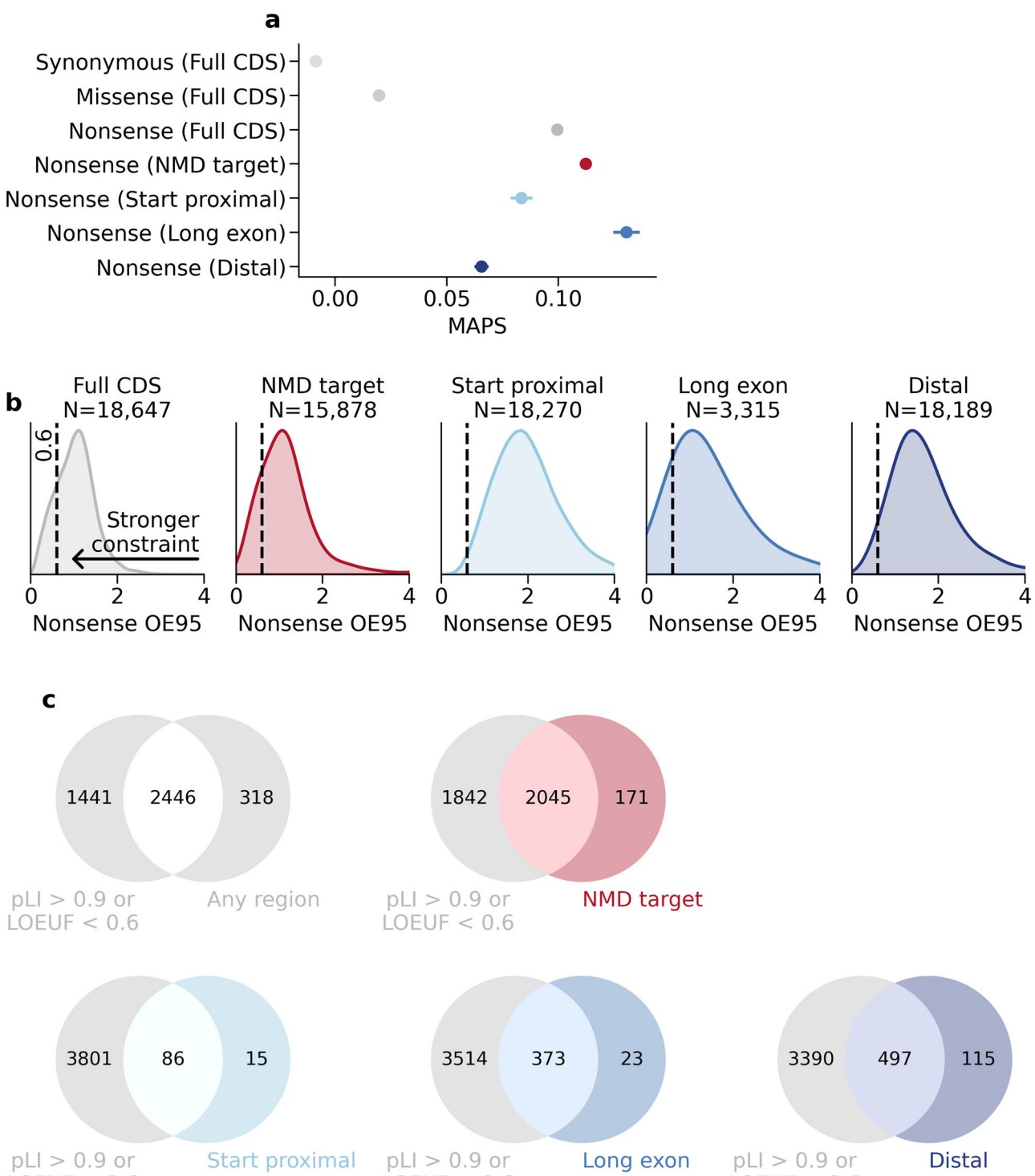

**Fig. 3 | Regional nonsense constraint identifies additional transcripts which are intolerant to PTCs. a** MAPS for nonsense variants in NMD-target regions (red) and NMD-escape regions (blue). Error bars show 95% confidence intervals. All categories are significantly different from one another (pairwise two-sided proportions Z tests with Bonferroni correction for 21 tests at alpha=0.05, P < 0.0024, asterisks not shown). **b** Distributions of the upper bounds of the 95% confidence interval around the observed/expected ratio of nonsense variants (nonsense OE95), stratified by NMD region. The dashed black line marks OE95 = 0.6. The number of transcripts in which the nonsense OE95 could be quantified is shown. **c** Venn diagrams showing the intersection between transcripts with regional nonsense constraint, and transcript-level constraint in gnomAD v4.1 (pLI > 0.9 or LOEUF < 0.6). Source data are provided as a Source Data file.

significantly higher odds of diagnosis (OR 3.89 (3.18–4.76), 5.85 (3.96–8.66), 5.75 (3.96–8.37) respectively). Importantly, the odds of diagnosis are unchanged for carriers of de novo nonsense and frameshift variants in unconstrained regions (Fig. 6a). Diagnostic outcomes for carriers of de novo nonsense and frameshift variants

therefore strongly align with regional constraint, highlighting the clinical utility of this metric for variant prioritisation and interpretation in the molecular diagnosis of rare genetic conditions.

Next, we explored a potential role for regional nonsense constraint to improve clinical variant interpretation using classifications

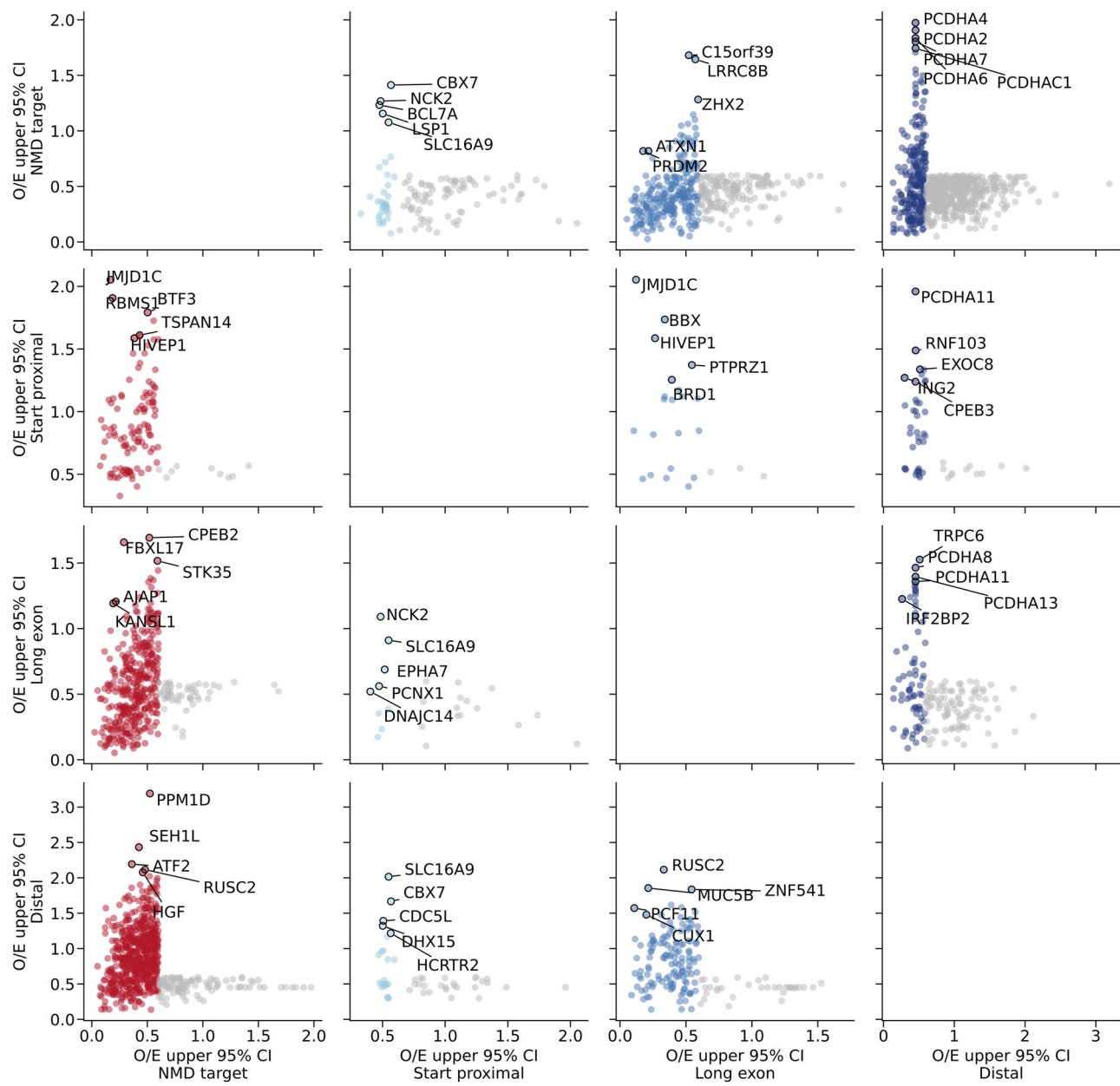

**Fig. 4 | Non-uniform nonsense constraint within transcripts.** Each panel compares nonsense OE95 between NMD regions in the same transcript. Transcripts are only plotted if at least one of the regions is constrained. Points are coloured according to the region on the horizontal axis; NMD-target regions are shown in red, NMD-escape regions in blue. Points in grey depict transcripts that are not constrained in that region. Within each panel, the five transcripts with the greatest absolute difference in nonsense OE95 scores are labelled.

for 1,541,121 variants in ClinVar[25]. In known autosomal dominant disease genes, we find that (likely) pathogenic nonsense and frameshift variants are more than twice as likely to fall in constrained regions than unconstrained regions (Supplementary Fig. 18). Conversely, variants of uncertain significance (VUS) and (likely) benign variants are up to four times more likely to fall in unconstrained regions. This trend is not observed for autosomal recessive disease genes (Supplementary Fig. 19). These results suggest that our metric has the potential to inform variant classification in monoallelic disorders caused by PTCs.

### Constrained regions are enriched for de novo nonsense and frameshift variants in rare disease trios

Selective constraint can be used to prioritise novel causative genes in monogenic human diseases, including developmental disorders[22,44].

Half of severe developmental disorders are caused by de novo variants[45] and analysis of de novo variation in large cohorts has successfully identified novel disease genes and damaging functional variant classes[46,47]. To explore the contribution of PTCs in constrained regions to monogenic disorders, we compared the abundance of de novo nonsense and frameshift variants in constrained versus unconstrained regions in a combined dataset of three large cohorts of rare disease trios (32,260 trios in total). Here, the NMD region of a frameshift variant was determined by the position of the downstream PTC. After adjusting for sequence length and mutation rate, we find a substantial enrichment of de novo nonsense and frameshift variants in constrained transcripts (5.07-fold enrichment) (Fig. 6b). Strikingly, this is more pronounced in constrained long exon and distal regions (6.11- and 9.52-fold enrichment, respectively) than NMD-target regions (4.38-fold enrichment).

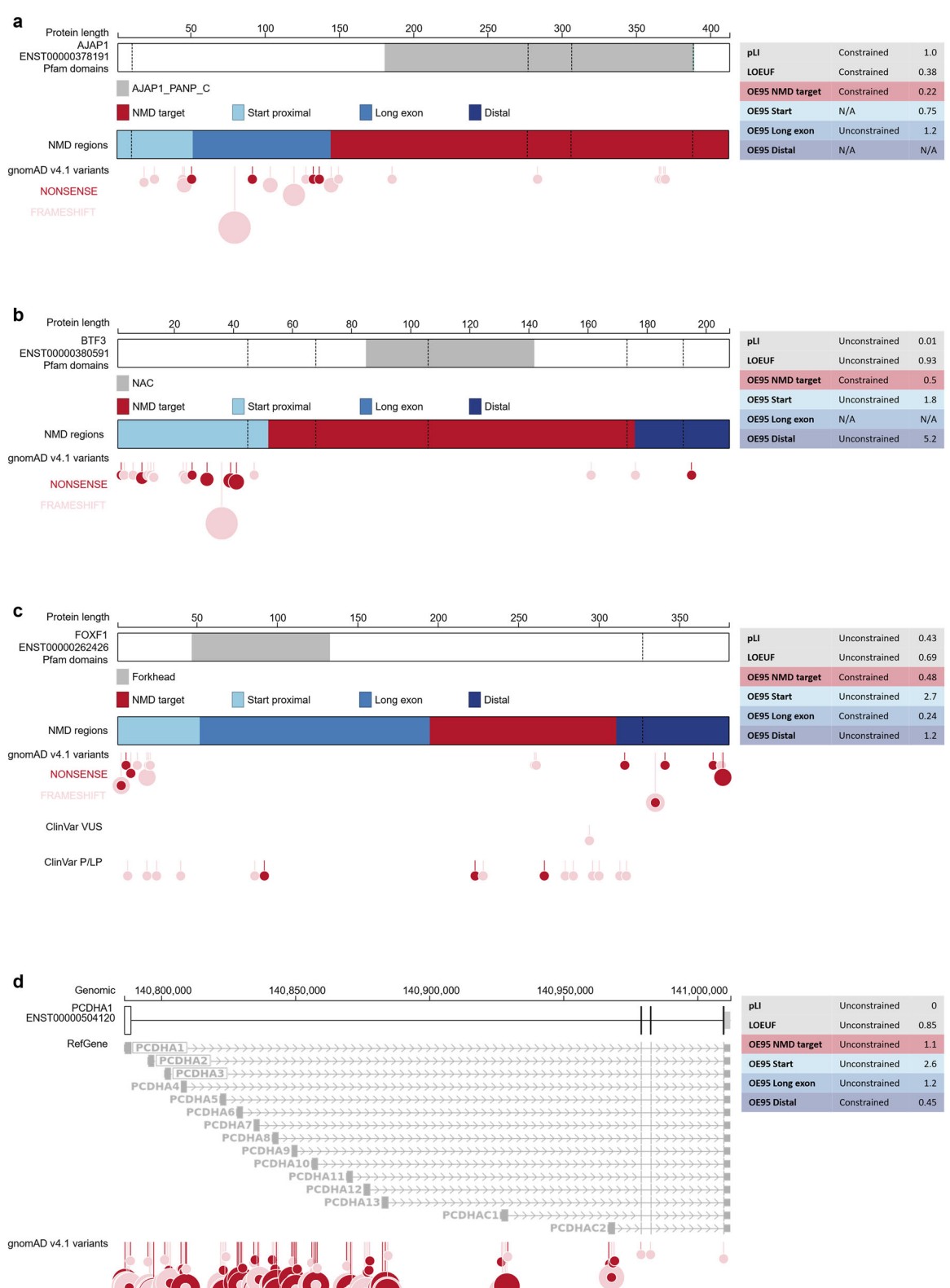

We subset our analysis by disease annotations in OMIM[27]. De novo nonsense and frameshift variants are highly enriched in constrained regions of known dominant disorder genes (Fig. 6c), but not recessive disorder genes (Fig. 6d). Importantly, this regional enrichment persists in genes with no disease annotation in OMIM (Fig. 6e), implying that these variants are an indicator for novel disease genes that remain to be discovered.

## Clustering of de novo variants in constrained regions highlights 22 candidate disease genes

To explore novel candidate disease gene associations, we curated genes with clusters of de novo nonsense and frameshift variants in constrained regions in our dataset. In total, we identified 999 genes with at least one de novo nonsense or frameshift variant in a constrained region. Of these, we find 251 genes with three or more such

**Fig. 5 | (overleaf)| Many transcripts have non-uniform nonsense constraint.**
**a** Nonsense variants are not constrained in the long exon NMD-escape region of
*AJAP1* (ENST00000378191). The CDS for the MANE Select transcript is shown (thick
boxes). Exon junctions are marked by vertical dashed lines. Pfam domains are
shown in grey (top); the short name of the domain is given in the legend. NMD-
target regions (red) and NMD-escape regions (blue) are shown (middle). Nonsense
variants (red circles) and frameshift variants (pink circles) from gnomAD v4.1 are
shown (bottom). The size of the circle corresponds to the allele count of the variant.
Regional nonsense OE95, pLI and LOEUF scores are shown (right). Note that

ENST00000378191 contains a 3' UTR intron, and therefore the CDS does not
contain a distal NMD-escape region. **b** *BTF3* (ENST00000380591) is constrained for
nonsense variants in the NMD-target region. **c** *FOXF1* (ENST00000262426) non-
sense and frameshift variants are bimodally distribution in gnomAD v4.1. Patho-
genic nonsense variants in ClinVar are more centrally located. **d** *PCDHA1*
(ENST00000504120) and other protein-coding genes in the protocadherin alpha
cluster (chr5:140786140-141012347) have strong nonsense constraint in their distal
NMD-escape regions. The canonical transcript for every protein-coding gene in the
cluster is shown in grey.

variants in a constrained region, including 57 genes which have no
dominant monogenic disease association in OMIM at present (Fig. 6f,
g, Supplementary Data 2). We propose that this gene set is likely to
include novel monogenic disorder genes. Indeed, several of these
genes have recently been associated with developmental disorders
(e.g. *FOSL2*[12], *PCBP2*[48], *ZFHX3*[49]). After excluding genes reported in
disease gene curation databases (PanelApp Australia[50] and
Gene2Phenotype[51]) and filtering for regions with nonsense O/E
values < 0.3 and fewer than 20 nonsense variants in gnomAD v4.1, we
highlight a set of 22 genes (*ADGRB1, ARID4A, ARID5B, CLASP1, FBRS,
FIGN, GPBP1, HDAC2, HDLBP, MAGI2, MGA, NAA25, NCL, PSMD6,
RANBP9, RIF1, RLF, TLE3, UBR5, VEZF1, WAPL* and *ZSWIM8*) as potential
candidates for novel disorders. Taken together, these results show that
PTC-introducing de novo variants in rare disease cohorts are enriched
in both NMD-target regions and predicted NMD-escape regions, and
that novel disorders driven by heterozygous variants in these regions
remain to be discovered.

## Discussion

NMD is critical for determining the molecular consequences of variants
which introduce PTCs, including nonsense, frameshift, and splicing
variants. Whereas PTCs predicted to trigger NMD are likely to cause
LoF, those predicted to escape NMD may give rise to truncated protein
products with hypomorphic, gain-of-function or dominant-negative
effects. NMD efficiency is largely determined by the position of the PTC
within a transcript; PTCs in the first 150 nt of the coding sequence, in
the last exon, in the last 50–55 nt of the penultimate exon, or in the 5'
end of extremely long exons (>~400nt), are predicted to escape NMD.
Although these predicted NMD-escape regions span nearly 40% of the
coding genome (Fig. 1c), current pLoF constraint metrics, including
LOEUF[16] and GeneBayes[18] do not specifically account for NMD.

Here, we sought to quantify regional constraints against nonsense
variants to give clinical and biological insights into the impact of PTCs
in genetic disease. Previous work has examined regional constraints
predominantly against missense variants[19,20,52]. The number of possible
and observed nonsense variants is substantially less than for missense
changes, and our analyses are only possible because of the scale of
population sequencing resources now available[16] and the sophistica-
tion and accuracy of newer mutational models[26]. Nevertheless, we have
limited power to detect constraint in small genomic regions. For
example, we identify only 101 transcripts which are constrained in
start-proximal regions, which we define to have a maximum length of
150 nt. Increasing the size and diversity of sequencing cohorts and the
sharing of data between institutions will maximise the power of these
analyses to identify signals with finer resolution.

We focus on nonsense variants specifically, to contrast the con-
sequences of protein-truncation (in predicted NMD-escape regions)
and loss of function (predicted NMD-target regions), and to comple-
ment previous studies of protein-altering variation[19,20]. Importantly, we
did not include indels in our constraint model because of the challenge
of estimating mutation rates for these variants, and because frame-
shifted sequences can have distinct pathomechanisms to PTCs alone[53].
For example, although heterozygous frameshift variants in the
penultimate exon of *DVL1* cause autosomal dominant Robinow syn-
drome (MIM 616331), *DVL1* is not constrained for nonsense variants in

any region, because the pathomechanism for this disorder is depen-
dent on gain of function by a specific frameshifted sequence[54].

We also intentionally used a biologically informed, rules-based
approach to define our NMD regions[23], which we expect to offer
clearer biological insight and have greater relevance to clinical variant
interpretation frameworks. However, regional constraint metrics
which are agnostic to the positional 'NMD rules' may offer different
perspectives about the regional vulnerabilities of genes to PTCs. For
example, a recent study identifying variant clusters using Gaussian-
mixture models identified non-uniform distributions of pLoF variants
in Mendelian disease genes in the UK Biobank, explaining reduced
penetrance in unaffected carriers of these variants[35].

We find that the strength of selection against PTCs differs across
NMD regions (Fig. 2a). NMD-escaping variants are generally under
weaker negative selection than NMD-targeted variants, consistent with
previous studies[6]. These results suggest that for most genes, complete
loss of one allele is more deleterious than truncation at the 5' or 3' gene
termini. However, nonsense variants in long exon regions are more
challenging to interpret. Whereas variants here are more highly con-
strained than in any other NMD region (Fig. 3a) and PTVs in long exon
regions are over-represented and depleted for VUS in ClinVar (Fig. 1d,
e), yet long exon regions are the least likely to contain a Pfam domain
(Supplementary Fig. 8) and have the weakest phyloP, CADD, and
AlphaMissense scores of any region (Supplementary Fig. 7). Impor-
tantly, NMD efficiency is generally greater in long exon regions than
either start-proximal or distal regions (Fig. 2). Our interpretation of
these data is that PTCs in long exon regions, even if they do escape
NMD, will nevertheless truncate a large proportion of the CDS.
Mechanistically, therefore, they are more likely to result in loss of
function. As a group, genes with unusually long exons may also be
generally important for reproductive fitness.

At the transcript level, we found 2764 transcripts with significant
regional constraint using a new OE95 metric. Interestingly, 12–19% of
these are currently not annotated as constrained using existing
transcript-level metrics such as LOEUF[16] and GeneBayes[18]. Importantly,
we find hundreds of transcripts in which constraint is not uniform
across NMD regions. For example, 755/2216 transcripts (34%) con-
strained in NMD-target regions are not constrained across the full
transcript (Supplementary Fig. 3). We highlight several examples with
biological and clinical salience (Fig. 5), including monogenic disease
genes with known regional intolerance to PTCs, such as *GATA6* (Sup-
plementary Fig. 1) and *ASXL3* (Supplementary Fig. 8).

We also used these data for translational insights by jointly ana-
lysing de novo variation across three large rare disease cohorts. For
example, we found that the odds of receiving a diagnosis for a
monogenic disorder are up to 5.9-fold higher in individuals with de
novo variants in constrained regions, but unchanged for variants in
unconstrained regions (Fig. 6a). In principle, these data could be
incorporated into variant interpretation guidance under a Bayesian
framework[55]. Current ACMG guidelines only consider NMD-escape in
the context of distal PTCs[21]. The strength of evidence for pathogenicity
in these cases is determined by the loss of a functional protein domain,
or loss of >10% of the protein. Our metric improves on these heuristics
by offering a quantitative, continuous, and per-gene statistic for
regional sensitivity to PTCs in distal, long-exon, and start-proximal

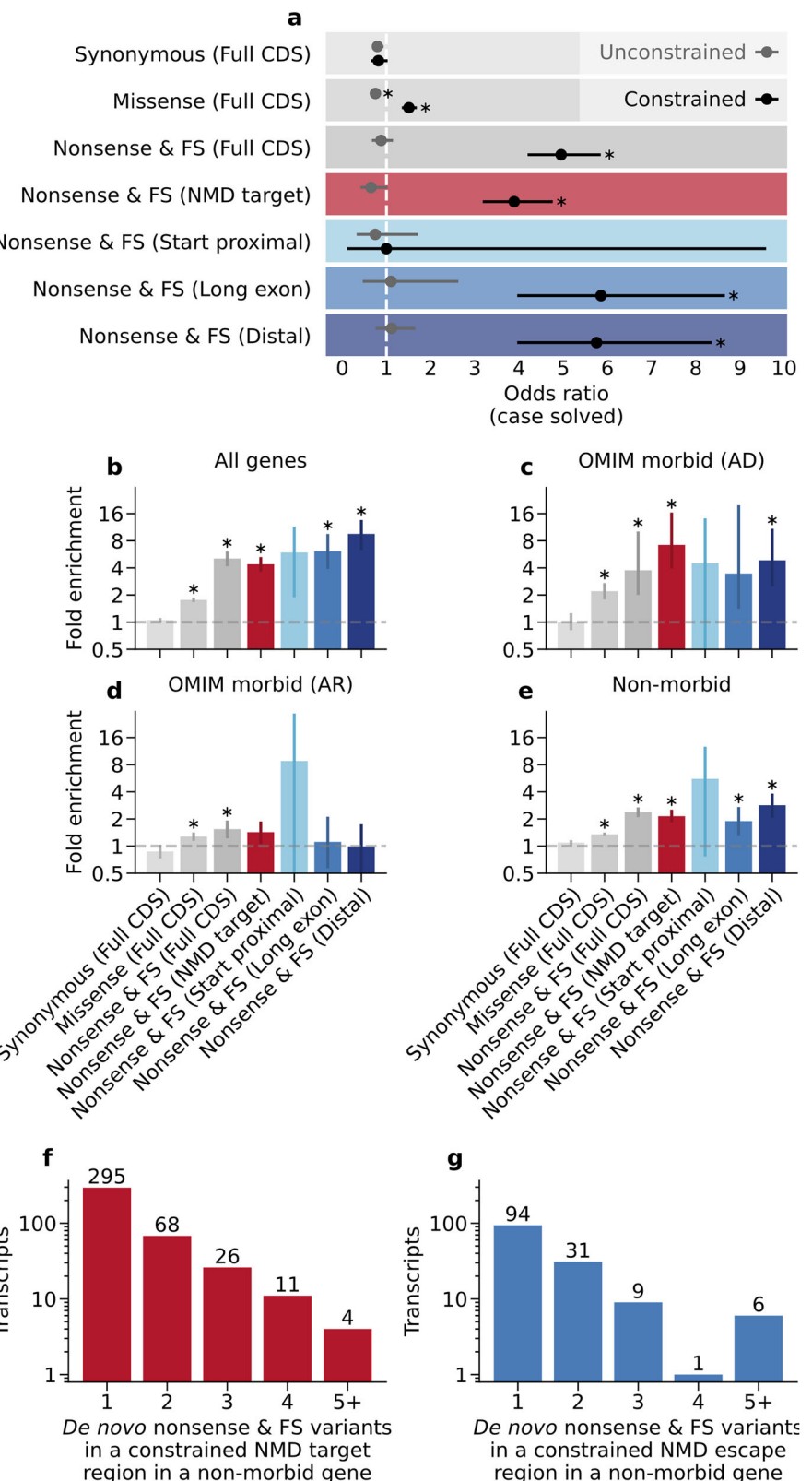

regions. We propose that our metric will complement the ACMG guidance and aid clinical judgement in the classification of PTCs. Because most truncating VUS in ClinVar fall in unconstrained regions (Supplementary Fig. 19), our constraint metric may also be used to inform benign, as well as pathogenic, variant classifications.

Constraint metrics have previously been used to prioritise novel candidate disease genes[19], and others have used de novo variant enrichment in 3' gene termini[4,36] and the whole CDS[47] to identify new disease genes. By defining NMD status across the full CDS, we could explore distributions of de novo variants at higher resolution throughout the gene. Broadly, we observe that de novo variants are enriched in constrained regions throughout the CDS (Fig. 6b). The persistence of this enrichment in genes with no known disease association in OMIM[27] (Fig. 6e), suggests significant opportunities for novel

**Fig. 6 | Regional nonsense constraint highlights clinically impactful de novo variants and novel disease gene candidates. a** Odds ratio for receiving a diagnosis ('case solved') for 12,303 coding de novo variants in 13,908 rare disease trios in the 100,000 Genomes Project. Error bars represent 95% confidence intervals. Asterisks indicate statistically significant values (two-sided Z test on log-transformed values with Bonferroni correction for 14 tests at alpha=0.05, p < 0.0036). **b–e** Relative enrichment of coding de novo variants in constrained versus unconstrained regions for 50,825 variants in 32,260 rare disease trios. Relative enrichment is shown for all genes (**b**), monogenic disease genes associated with autosomal

dominant phenotypes in OMIM (**c**), monogenic disease genes associated with autosomal recessive phenotypes in OMIM (**d**) and genes with no monogenic disease association in OMIM (**e**). Error bars show bootstrap 95% confidence intervals. Asterisks show statistically significant values (two-sided percentile bootstrap with Bonferroni correction for 28 tests at alpha=0.05, p < 0.0018). The number of canonical transcripts with a constrained NMD-target region (**f**) or NMD-escape region (**g**) harbouring one or more de novo nonsense or frameshift variants in 32,260 rare disease trios. FS = frameshift variant. Exact P values for **a**, **b**, **c**, **d**, and **e** are provided in the Source Data file.

disease gene discovery. By identifying clusters of de novo variants in constrained regions, we prioritise 22 genes as potential candidates for novel disorders.

In conclusion, we apply mutational models to biobank-scale sequencing data to construct a regional nonsense constraint metric offering biological and clinical insights into human genetic disease. We find over 2000 genes with regional nonsense constraint, show that de novo variants and diagnostic outcomes align with constraint in disease cohorts, and prioritise 22 candidate disease genes. We suggest that this NMD-aware nonsense constraint annotation can facilitate clinical interpretation of PTCs, drive novel disease-gene discoveries, and will be of considerable utility to studies of pLoF variation in health and disease.

## Methods

### Representative transcripts

One representative transcript was selected for every protein-coding gene in the GENCODE v39 comprehensive gene annotation[56]. Transcripts were filtered using the following criteria: 'tag contains 'Ensembl_canonical', 'gene_type == 'protein_coding', 'transcript_type == 'protein_coding'. 19,676 transcripts meeting these criteria were identified. Transcripts were not included or excluded on any other criteria, including transcript length or structure. Where available, the MANE Select transcript (MANE v0.95)[57] was chosen ($N$ = 18,584). For genes lacking a MANE annotation, the APPRIS principal isoform[58] was chosen ($N$ = 1092).

### Defining NMD regions

For each transcript, the following predicted NMD regions were annotated with custom scripts:

- Start-proximal: CDS positions <=150nt downstream of the translation start site.
- Long exon: CDS positions >400nt upstream of an exon-intron junction.
- 50nt rule: CDS positions <=50nt upstream of an exon-intron junction in the penultimate exon (the second-most 3' exon).
- Last exon: CDS positions in the final exon (the most 3' exon).
- NMD-target: CDS positions lacking any of the above annotations.

The 'last exon' and '50nt rule' annotations also account for transcripts which contain 3' UTR introns, which comprise ~4% of canonical transcripts. For example, in these transcripts the most 3' CDS positions do not have a 'last exon' annotation. 'Last exon' and '50nt rule' annotations were unified into one category: 'Distal'. Positions with multiple NMD region annotations were assigned one definitive annotation with this priority: start-proximal > distal > long exon. Transcripts which lacked any of these regions were retained throughout the analysis; no constraint annotations were provided for the absent regions.

### Consequence annotation of coding SNVs

The set of all possible single-nucleotide variants (SNVs) in our CDS of interest was annotated with custom scripts using bedtools v2.31.1[59], BCFtools v1.20[60], and the GRCh38 reference genome (GenBank assembly accession GCA_000001405.28). SNVs were annotated with the Variant Effect Predictor (VEP) v105[61] and the Ensembl v105 cache,

which is consistent with MANE v0.95[57] and GENCODE v39[56]. Only autosomal SNVs annotated as 'synonymous_variant', 'missense_variant', or 'stop_gained' in the Ensembl canonical transcript for each gene were kept, yielding 22,563,474 synonymous, 71,502,748 missense, and 4,047,794 nonsense SNVs.

### Allele-specific expression analysis

100KGP participants with RNA-Seq data were identified from the 'transcriptome_file_paths_and_types' table in the Genomics England LabKey application (release v19). Where participants had more than one RNA sample, the most recently sequenced sample was chosen. In total, RNA-Seq data were available for 7829 participants. Of these, 5132 individuals had paired WGS data available in an aggregated variant call set from 78,195 participants in the 100KGP (AggV2). Nonsense variants in aggV2 were identified through annotation with the Ensembl Variant Effect Predictor (VEP v105)[61]. Only autosomal variants with a 'stop_gained' consequence in an Ensembl canonical transcript were retained. 100,839 unique nonsense variants with FILTER = 'PASS' were identified in AggV2. The following filters were applied to these variants: variants with AC > = 1 in 5132 samples with paired RNA-Seq data ($N$ = 16,975); variants at biallelic sites only ($N$ = 13,241).

Synonymous variants were treated identically, yielding 352,031 variants. To enable comparisons between nonsense variants and synonymous variants with similar expression levels and rarity, synonymous variants were further filtered for singleton alleles occurring in a gene in which at least one nonsense variant had been identified above ($N$ = 95,058). This set of synonymous variants was then randomly down-sampled to match the number of nonsense variants ($N$ = 13,241).

Both nonsense and synonymous variants were further filtered to those at heterozygous sites only, passing all sample-level filters (FMT/FT = 'PASS'), and with variant allele fractions between >=0.25 and <=0.75. 12,798 nonsense variants and 12,823 synonymous variants meeting these criteria were identified. All 5132 samples contained at least one qualifying nonsense variant.

Allele-specific expression (ASE) analysis was performed with the Genome Analysis Toolkit ASEReadCounter (GATK v4.5.0.0)[62]. The analysis was limited to the variants described above, using default read filters. The following additional filters were applied: minimum read mapping quality >= 20, minimum base quality >= 20, minimum depth of non-filtered base >= 10. ASE was quantified for 5203 unique synonymous variants and 4855 unique nonsense variants in total.

NMD efficiency was inferred from the proportion of RNA-Seq reads carrying the variant allele (variant allele fraction, VAF). In order that the results were not skewed by recurrent variants, or variants which overlapped more than one gene, the analysis was limited to singleton alleles which overlapped exactly one gene, totalling 5120 synonymous variants and 3750 nonsense variants. VAFs between variant classes were compared with two-sided Mann-Whitney U tests.

### SNVs in gnomAD v4.1

SNVs were identified in exome sequencing data from gnomAD v4.1[63]. Variants were filtered through the gnomAD sample and variant quality control (QC) pipeline described here: https://gnomad.broadinstitute.org/news/2023-11-gnomad-v4-0/. SNVs in our transcripts of interest, which passed all variant filters (FILTER = 'PASS') and had a minimum

allele count (AC) of 1, were extracted from the 730,947 samples which passed sample-level QC. In total, 17,885,374 SNVs meeting these criteria were identified. SNVs were labelled with the consequence annotations described above.

## Mutability-adjusted proportion of singletons

For CDS variants observed in gnomAD with median coverage >=20 ($N = 17,665,054$), we calculated the proportion of singletons (AC == 1) for each consequence (synonymous, missense, nonsense) and NMD region (NMD-target, Long exon, Distal, Start proximal) using custom scripts.

Variants in more mutable sequence contexts are more common and have a lower proportion of singletons. To account for sequence mutability, we fit a weighted least squares model of the proportion of singletons against the scaled mutation rate for synonymous variants in each context. We used mutation rate estimates from Roulette[26] (see below). There were 99 unique values for the scaled mutation rate among the observed synonymous variants. The model was weighted by the number of possible variants in each variant context.

For each functional variant class, we used the model to predict the expected proportion of singletons. This number was subtracted from the observed proportion of singletons to find MAPS.

For statistical comparison of MAPS between each variant class, all MAPS scores were adjusted by the expected proportion of singletons for synonymous variants before performing pairwise two-sided Z tests of proportion with Bonferroni correction.

## Mutational model for coding SNVs

The expected number of variants among our SNVs of interest was derived from mutation rate estimates from Roulette[26]. Roulette is a base-pair-level mutation rate model which accounts for local sequence context, DNA methylation, and transcription to accurately model SNV mutation rates genome-wide[26]. Mutation rate estimates from Roulette were scaled to the gnomAD v4.1 exomes to account for differences in cohort size and composition, as described here: https://github.com/vseplyarskiy/Roulette/tree/main/adding_mutation_rate. The set of high-quality synonymous sites described in Seplyarskiy et al.[26] were used as 'background' or 'neutral' variants for calibrating mutation rate estimates ($N = 19,969,231$). Synonymous variants from the background set were identified among the 17,885,374 gnomAD-observed SNVs described above. Scaled mutation rate estimates were calculated for all possible SNVs genome-wide. We filtered for SNVs in our CDS positions of interest, yielding annotations for 95,668,203 unique SNVs. The scaled mutation rates provide an estimate of the probability of observing each SNV in the gnomAD v4.1 exomes[16], in the absence of negative selection.

## Regional nonsense constraint

Because transcripts and regions which are poorly covered may appear depleted of nonsense variants (false positives), we only included positions with median sequencing depth >=20 in this analysis.

The proportion of variants expected in each transcript and NMD region was taken as the mean scaled mutation rate for each functional variant class (synonymous, missense, and nonsense variants) in each region. We multiplied this proportion by the number of possible variants, to obtain the number of expected variants in each region. This approach accounted for mutability, the number of possible variants of each class, and the sequence length of each region.

For each transcript, region, and functional variant class, we treated expected variant counts as a Poisson binomial random variable with the number of trials ($N$) equal to the number of possible variants, and the probability of observing each variant ($P$) equal to the scaled mutation rate of that variant under the null model (above).

To identify constrained transcripts and regions, we performed a one-sided test for the number of variants observed against the given Poisson binomial distribution[64]. We tested the null hypothesis that the number of observed nonsense variants is equal to or greater than the number of expected nonsense variants in each transcript and region.

We also calculated the ratio of observed/expected variants (O/E), as well as the upper bound of the 95% confidence interval for the O/E value (OE95). We defined OE95 as the upper bound of the 95% binomial confidence interval for the proportion of variants observed, multiplied by the number of possible variants and divided by the number of expected variants.

After correcting for multiple testing with the Benjamini-Hochberg/false discovery rate (FDR) method[65], we defined constrained transcripts and regions as those meeting the following criteria:

- Fewer nonsense variants than expected (one-tailed binomial test, FDR < 0.05)
- The number of synonymous variants observed is not nominally lower than the number expected (one-sided binomial test, $P > 0.05$)
- The upper bound of the 95% confidence interval of the nonsense O/E value (nonsense OE95) is less than 0.6.

Unconstrained transcripts and regions were defined as follows:
- Non-significant constraint scores prior to FDR correction (one-tailed Z test, $P > 0.05$).
- At least one nonsense variant observed in the cohort.

Transcripts and regions which did not meet the criteria for "constrained" or "unconstrained" status were annotated as "indeterminate".

## phyloP, CADD, AlphaMissense and pext annotations

Base-level phyloP scores from multiple sequence alignments of 241 mammals[28], Combined Annotation Dependent Depletion (CADD) scores[30], AlphaMissense scores[29], and base-level proportion expressed across transcripts (pext) scores[31] were downloaded from their publicly available repositories. Download URLs are given in the *src/downloads/* directory in the accompanying code repository. CDS sites and variants of interest were annotated with these scores using custom scripts. Site-level AlphaMissense scores were taken as the lowest AlphaMissense score per site per transcript. We successfully annotated 33,834,625 CDS sites with phyloP scores, 93,913,916 CDS SNVs with CADD, 19,900,876 CDS sites with AlphaMissense, and 32,120,976 CDS sites with pext scores.

## Intersection of constrained regions with Pfam domains

High-confidence, manually curated Pfam-A domains overlapping GENCODE v47 genes were downloaded in full from the 'Pfam domains in GENCODE genes' track of the UCSC table browser[66]. Identically-named Pfam domains with overlapping coordinates and strand were merged into contiguous features using BEDtools v2.31.1[59]. These features were then intersected with our NMD regions. The proportions of constrained and unconstrained regions overlapping a Pfam domain were calculated and compared with two-tailed, two-sample Z tests of proportion with Bonferroni correction.

## Truncating variants in ClinVar

Clinically interpreted variants were obtained from the ClinVar[25] variant_summary.txt file downloaded on 14/11/23. We filtered variants on the following criteria: Assembly == GRCh38; autosomal contigs (chr1-22) only; presence of a valid RefSeq ID; reference and alternate alleles present; ReviewStatus does not contain either of 'no assertion' or 'no interpretation'; ClinicalSignificance does not contain any of 'not provided', 'drug response', 'other', 'risk', 'low penetrance', 'conflicting', 'affects', 'association', 'protective', 'confers sensitivity'; no conflicting ACMG classification. We identified 2,022,542 unique variants meeting these criteria. These were further subset to variants which fell in the MANE_Select transcript of the respective gene (MANE v0.95, $N = 1,997,924$).

We annotated these variants with VEP v105[61] and the Ensembl v105 cache. Only autosomal SNVs annotated as 'synonymous_variant', 'missense_variant', 'stop_gained', or 'frameshift_variant' in the Ensembl canonical transcript for each gene (where the HGNC[67] ID of the gene matched the ClinVar annotation) were kept. We successfully annotated 1,542,731 variants. Of these, 1,541,121 variants were within a region with a valid regional nonsense constraint annotation.

## Known disease genes in OMIM

A list of known monogenic disease genes was obtained from the OMIM[27] genemap2.txt file downloaded on 06/11/23. Genes were excluded from the morbid gene list if they had no associated phenotype in OMIM, no mode of inheritance was given for the disorder, no Ensembl gene identifier was available, or if the gene-disease association was annotated as 'non-disease', 'susceptibility', or 'provisional'. In total, 3985 monogenic disease genes were identified.

## De novo variants

De novo variants (DNVs) overlapping our transcripts of interest were identified in short-read sequencing data from multiple cohorts of rare disease trios.

First, we used a set of high-confidence DNVs from whole genome sequencing (WGS) of 13,932 trios in the 100,000 Genomes Project (100KGP)[42]. The annotation pipeline used to identify these variants is publicly available at this URL: https://re-docs.genomicsengland.co.uk/de_novo_data/. We excluded duplicate entries and cases with no identified de novo variants, retaining 13,908 trios in total. For 1905 of these trios, variants were annotated against GRCh37. Variants for these participants were lifted over to GRCh38 using Picard (version 3.1.1)[68]. In total, we identified 979,640 variants in 13,908 individuals.

Second, we used a set of publicly available coding DNVs from exome sequencing of 31,058 rare disease trios[47]. This cohort was assembled through a collaboration between the Deciphering Developmental Disorders Study (DDD)[69], Radbound University Medical Center (RUMC), and GeneDx. These data consist of 45,221 high confidence de novo calls in 23,902 individuals, annotated against GRCh37. These variants were lifted over to GRCh38 as described above, yielding 45,215 variants in 23,900 individuals.

Finally, we extracted DNVs from whole-genome sequencing data in 29,872 participants in the rare disease arm of the NHS Genomic Medicine Service (GMS). Details of the alignment, QC, and variant calling pipeline for this cohort are publicly available here: https://re-docs.genomicsengland.co.uk/rare_disease_2_2.pdf. We filtered DNVs from 14,260 joint-called gVCFs using the following criteria: FILTER == 'PASS' in all samples, allele count == 1, de novo flag == 'DeNovo', de novo quality score >= 0.458, 0.25 <allele fraction <0.75, and no more than one read supporting the alternate allele in either parent. In total, we found 462,473 qualifying DNVs in 5925 trios.

DNVs from all cohorts were concatenated with BCFtools[60], and annotated with allele frequencies from exome sequencing data in gnomAD v3.1.1[16] and aggregated variant calls from whole genome sequencing of 78,195 individuals in the 100KGP or 29,872 individuals in the GMS (see below). Variants were then annotated with VEP (v105)[61]. Finally, annotated DNVs were filtered on the following criteria:

- Allele count in gnomAD v3.1.1 exomes <= 1
- Allele frequency in gnomAD v3.1.1 exomes <= 0.0001
- Allele count in 100KGP < = 5 (100KGP variants only)
- Allele count in GMS < = 5 (GMS variants only)
- VEP consequence annotation of 'synonymous_variant', 'missense_variant', 'stop_gained', or 'frameshift_variant' in the Ensembl canonical transcript of a gene.

A small number of families are likely to have participated in more than one of the DDD, 100KGP, and GMS studies. To conservatively mitigate against 'double counting' of DNVs from these individuals, we excluded from the DDD set any DNV which was also observed within the 100KGP set or the GMS set. 1,294 DNVs were excluded from the DDD set in this manner. Similarly, we excluded from the GMS set any DNV which was also observed within the 100KGP set. 172 DNVs were excluded from the GMS set in this manner. We judged that participant duplication between the other cohorts was highly unlikely.

In total, 52,479 de novo variants (49,793 unique by chromosome, position, reference allele, and alternate allele) meeting these criteria were retained in 32,260 trios.

These variants were annotated according to the NMD region in which they fell (determined by the variant position in the VCF file), and with regional nonsense constraint labels. NMD regions were assigned to 51,984 DNVs. The remaining 513 DNVs are indels where the variant position is outside the CDS. These variants were excluded from downstream analyses. 50,067 DNVs fell in a region in which nonsense constraint could be quantified, of which 31,892 DNVs fell in a region with an unambiguous nonsense constraint annotation (constrained $N = 14{,}568$, unconstrained $N = 17{,}324$).

## Annotation of de novo frameshift variants with Aenmd

Frameshift variants commonly introduce a PTC downstream of the indel. Therefore, to include frameshifts in our analyses of de novo variants, we annotated NMD regions according to the position of the downstream PTC using the NMD-escape predictor Aenmd[23] with default parameters against the GENCODE v43 annotation. Of 4108 unique de novo frameshift variants, 3912 variants (95.2%) were successfully annotated with Aenmd. Variants with more than one sequence ontology term (e.g. 'frameshift_variant&start_lost') were excluded to limit to unambiguous frameshifts, yielding 3903 variants. Variants were filtered to those with an aenmd annotation in the Ensembl canonical transcript for that gene, yielding 3895 variants. Of these, an HGVS p. identifier could not be determined in one variant, and the frameshift extended into the 3' UTR in seven variants. In total, the location of the PTC was annotated in 3887 variants (94.6% of the unique de novo frameshifts in the cohort). Among these, the NMD region differed from our original annotation for 660 variants (17.0%) and was unchanged for 3227 variants (83.0%).

## De novo variant enrichment

To quantify the regional enrichment of DNVs, we counted DNVs in constrained and unconstrained NMD regions. To enable direct comparisons, the absolute number of DNVs in each region was adjusted by the number of unique SNVs expected by our models in gnomAD v4.1 (above). For example, the number of unique synonymous DNVs in constrained distal regions was adjusted by the number of unique synonymous variants expected in those same regions in gnomAD. Missense variants were treated identically. Stopgain and frameshift variants were combined and adjusted for the number of nonsense variants expected in gnomAD. Finally, the adjusted number of variants in constrained and unconstrained regions was normalised by the adjusted number of variants in unconstrained regions. We quantified relative enrichment as the ratio of these normalised variant counts in constrained and unconstrained regions. Confidence intervals and P values for these ratios were calculated using the percentile bootstrap.

## Clinical outcome data

DNVs from the 100KGP cohort were annotated with phenotypic and clinical outcome data from the LabKey application within the Genomics England secure research environment. Specifically, these data were accessed from the 'rare_disease_analysis', 'rare_disease_participant_phenotype', and 'gmc_exit_questionnaire' tables within LabKey. Diagnostic outcomes were classified as follows: a case was classed as solved where the 'case_solved' value in LabKey was 'yes'

or 'partially'; a case was classed as unsolved where the 'case_solved' value in LabKey was 'no', 'unknown' or absent.

## Figures

Figure 1a, b was created with BioRender (BioRender.com/g72m993). Transcript diagrams were created with ProteinPaint[70]. All other figures were created with Matplotlib[71].

## Statistics and reproducibility

The statistical methods used throughout this work are described above. No statistical methods were used to predetermine sample size and no data were excluded from the analyses. Randomisation and blinding were not applicable to the design of the experiments in this work.

## Reporting summary

Further information on research design is available in the Nature Portfolio Reporting Summary linked to this article.

## Data availability

Regional nonsense constraint annotations and summary statistics are available in Supplementary Data 3, and for download at https://github.com/alexblakes/regional_nonsense_constraint/blob/main/data/final/regional_nonsense_constraint.tsv. Regional constraint annotations are also publicly available through the DECIPHER portal (https://www.deciphergenomics.org/). Source data are provided with this paper. For Fig. 6, export of Source Data from the Genomics England Research Environment is not permitted. These data are available to registered users of the National Genomic Research Library at /re_gecip/shared_allGeCIPs/AlexBlakes/regional_nonsense_constraint/figure_6.tsv.

Data from the National Genomic Research Library (NGRL) used in this research are available within the secure Genomics England Research Environment. Access to NGRL data is restricted to adhere to consent requirements and protect participant privacy. Access to NGRL data is provided to approved researchers who are members of the Genomics England Research Network, subject to institutional access agreements and research project approval under participant-led governance. For more information on data access, visit: https://www.genomicsengland.co.uk/research. Source data are provided with this paper.

## Code availability

The analysis pipeline used to generate all results and figures is available at https://github.com/alexblakes/regional_nonsense_constraint. A separate repository for analyses undertaken in the GEL Secure Research Environment is available at https://github.com/alexblakes/gel_nmd_dnms.

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

## Acknowledgements

A.B. is supported by a Wellcome PhD Training Fellowship for Clinicians and the 4Ward North PhD Programme for Health Professionals (223521/Z/21/Z). N.W. is supported by a Sir Henry Dale Fellowship jointly funded by the Wellcome Trust and the Royal Society (grant no. 220134/Z/20/Z)

and a research prize from the Lister Institute. C.A.J. acknowledges support from MRC project grant MR/T017503/1. S.B. acknowledges the support of the NIHR Manchester Biomedical Research Centre. This study has been delivered through the National Institute for Health and Care Research (NIHR) Manchester Biomedical Research Centre (BRC) (NIHR203308). The views expressed are those of the author(s) and not necessarily those of Wellcome, the NIHR or the Department of Health and Social Care. The authors would like to acknowledge the assistance given by Research IT and the use of the Computational Shared Facility at The University of Manchester. We gratefully acknowledge the participants of the National Genomic Research Library (NGRL), whose contributions made this research possible. Secure access to the NGRL under project ID 589 was provided by Genomics England, which delivers the NGRL in partnership with NHS England, and is wholly owned by the UK Department of Health and Social Care. The NGRL contains participants' health data collected by the NHS as part of their care, along with samples and data from their participation in research, for which fully informed consent has been obtained. This includes genomic and clinical data provided through the NHS Genomic Medicine Service, as well as data obtained through research studies, including the 100,000 Genomes Project and the Generation Study, both of which are delivered in partnership with the NHS, and from other research cohorts involving external collaborators.

## Author contributions

A.B., J.E., and S.B. conceptualised the study. A.B. collected data, performed the analysis, and wrote the manuscript. J.E., S.B., C.J., and N.W. edited the manuscript. N.W. provided technical expertise. C.J., J.E., and S.B. supervised this work.

## Competing interests

N.W. receives research funding from Novo Nordisk. All other authors declare no conflicts of interest.
