## [Transparent Peer Review file · Nature Communications]

Regional nonsense constraint offers biological and clinical insights into genetic disease

Corresponding Author: Professor Siddharth Banka

Version 0:

Reviewer comments:

Reviewer #1

(Remarks to the Author)

Blakes et al have developed a regional nonsense constraint methodology that confirms some of our fundamental understandings of NMD after PTCs are introduced. Importantly, this new measure can be used to improve the interpretation of LoF variants in rare disease diagnostics. In addition, the authors highlight a major limitation in current measures of pLoF constraint such as LOEUF that are used extensively by the genetics community to interpret potential diagnostic variants in rare disease cases and to drive discovery of new disease-gene associations. These current measures treat the whole transcript as having an equal chance of harbouring a LoF variant following a PTC event but 40% of the coding genome is predicted to escape NMD. Their new metric is capable of more accurately identifying genes under constraint and they highlight 25 new candidate disease genes with compelling evidence from their new metric and clusters of de novo variants in rare disease cases from the 100,000 Genomes Project. This is an important study that will of interest to the whole rare disease community and I just have a few comments/suggestions.

1. They describe existing guidelines (ref 21) to accurately interpret PTCs using the ACMG guidelines and these already take into account NMD-target or escape regions. Some more discussion of how using their metric is better for diagnostic interpretation than just following these guidelines would be welcome or, better still, how their metric could be used to improve the ACMG guidelines.
2. There are likely to be existing false positive diagnoses where, historically, clinical geneticists may not have considered whether NMD escape was likely and they would not have had these new measures of regional constraint. For example, the standard 100,000 Genomes Project pipeline highlights tier 1 LoF variants in known disease genes related to a patient's conditions and many of these resulted in a diagnosis. Could they identify potential examples in the 100,000 Genomes Project data or give estimates or discuss the likelihood of their approach "fixing" false positive diagnoses and implications.
3. Some of the results around long-exon variants (Figure 1d) seemed to suggest they were more likely to be P/LP and were highly constrained and behaved more like NMD-target rather than NMD-escape variants? It was a bit unclear to me how to interpret such variants in the future and this could be clarified more in the discussion.
4. Related to this, much of the analysis relies, understandably, on Clinvar classification of LoF variants into P/LP or VUS but how reliable is this given that probably many would not have been validated for being true LoF variants?

(Remarks on code availability)

Reviewer #2

(Remarks to the Author)

Blakes and colleagues describes a new metric for interpreting genetic variants based on nonsense mediated mRNA decay (NMD) based on comprehensive population sequencing data. Using this metric, the authors identify ~3k genes that have regional constraint based on the prevalence of mutations, which they use to interpret trio data, clinVar, etc. The authors did a nice job of code/data sharing for their annotations. Overall, this seems to be a useful metric and is nicely executed. I have one major critique that should be addressed and a few minor points, but would suggest this work seems close to ready for publication.

1: The glaring hole in this study, in my view, is the lack of sufficient transcriptomics data that would verify the NMD mechanism as the putative driver. My understanding is that GEL has RNA for most individuals. Otherwise, 1000G would have paired RNA and DNA for every donor. My suggestion is that the authors scan the genomes from these data sources and identify mutations predicted to trigger NMD. Then, looking in the RNA-seq data, one could perform an allele-specific abundance analysis in RNA. If the metrics that they develop are correct, one should see a significant depletion of the NMD-triggering RNA.

Minor:

2: Lines 130-134 should be accompanied with statistical analyses

3: I don't understand the heterozygous only / non-homozygous limitation (line ~290). Could the authors clarify why this is the case and comment on the occurrence of homozygous-driven diseases (as they note in the introduction with heterozygous variant causes of disease).

(Remarks on code availability)

The code looks roughly complete but has a glaring lack of any sort of documentation to help an outside reader. The code resource should be revised before publication.

Reviewer #3

(Remarks to the Author)

This study by Blakes et al. investigates how the position of premature termination codons (PTCs) within genes influences their degradation by nonsense-mediated decay (NMD), a key cellular quality control mechanism. The authors show that ~one-third of the human coding genome is predicted to escape NMD, potentially producing truncated proteins that may cause disease via dominant-negative or gain-of-function effects. Using population-scale exome data (gnomAD v4.1) and a novel regional constraint metric (OE95), they identify 2,764 transcripts with significant intolerance to PTCs in specific transcript regions—some of which are missed by existing metrics like LOEUF. These constrained regions are enriched for de novo truncating variants in rare disease trios, with stronger enrichment correlating with higher diagnostic yield. The study highlights 25 genes as novel disease candidates based on clustered de novo PTCs in constrained regions. Overall, the findings refine the interpretation of pLoF variants, improve variant pathogenicity classification, and offer new insights into disease gene discovery.

Although the study provides important results in PTC-variant interpretation field, I have some major concerns.

1) Although the canonical rule is the most established rule for variability in NMD efficiency, the other two rules (start-proximal < 150 bp) and the long exon rules are not robustly validated especially for the germline variants. Lindeboom et al. (2016, 2019) papers put forward those two non-canonical rules mainly for somatic variants. Even for somatic variants, the NMD efficiency varies a lot within those regions, for instance for start-proximal PTCs that have in-frame downstream translation initiation site are more likely to be NMD escapees due to alternative translation initiation mechanism.

For the long-exon rule, NMD efficiency greatly varies based on the distance from PTC to the end of the transcript. So for these two rules, it is not optimal to consider every PTC in those regions are NMD-escapees especially for germline variants. The authors should deep dive into these rules carefully, at least for additional signals such as in-frame downstream translation initiation site or PTC distance to the exon end. In this way, they can be more informative.

2) Figure 1D – Are these the normalized counts retrieved from all of the ClinVar nonsense/frameshifts or just the pathogenic(P)/likely pathogenic (LP) ones? Could the authors clarify about that?

3) Although the start-proximal, long-exon and distal regions are only forming a smaller part of the gene/transcript (Figure 1c), how the genes are constrained in those regions are the genes mostly constrained overall according to pLI or LOEUF scores? It seems that the genes that they identified that are constrained in any region (~2.7K) are mostly identified as constrained by pLI or LOEUF. What could be expected that there are a substantial number of genes that are constrained in those small regions, but not constrained at all. Could the authors give an explanation to that?

4) Although this metric is only developed for nonsense variants, the authors also analyzed the enrichment of frameshift variants in rare disease trios (Figure 5). I think frameshift variants greatly differ in terms of their NMD escape region definition. So the analysis of frameshift variant using a metric developed for nonsense variants is misleading.

5) How about the normalization procedure by the gene length? The authors did do it by the normalization of synonymous variants but they should try it by the length of region as well and compare it.

6) I reviewed the website for the code but I think some parts in the code are missing or outdated. For instance, could not find how they calculated the observed/expected ratio for each gene/transcript.

7) The feature analysis of long-exon, distal sites and start-proximal sites could be more deeper including the protein structural domain analysis and some other potential protein features.

8) It is also not clear that the authors just focused on the transcripts with at least 150 bp and/or long exon. For most of the short transcripts, these will not be available. And for some of transcripts (<=2 exons), most of transcript will be NMD-escape. How did they evaluate those short transcripts? Did they exclude them?

(Remarks on code availability)

The code is a usable resource for the community but some parts in the code are missing or outdated. For instance, could not find how they calculated the observed/expected ratio for each gene/transcript.

Version 1:

Reviewer comments:

Reviewer #1

(Remarks to the Author)

I am satisfied that the authors have addressed my questions and added relevant new detail where required. They have also significantly improved the study based on the feedback from the other reviewers by incorporating transcriptomic analysis and fixing the code availability issues.

(Remarks on code availability)

Reviewer #2

(Remarks to the Author)

The revisions have addressed all of my questions. I have no major concerns remaining.

(Remarks on code availability)

Appropriate for publication.

Reviewer #3

(Remarks to the Author)

The reviewers addressed my questions in a very detailed manner.

I think their results are overall noteworthy to be published and will be of high significance to the variant interpretation field. But I have two more concerns to be addressed as you can see below:

1) Although the start-proximal rule and long-exon rules have been shown before and now by the authors (Figure 2) associated with less NMD-efficiency, we see some variability among variants in those regions. So when we call all those regions as NMD-escape, that can be misleading especially for long-exon regions. Also as you can see the results from the Figure 2, it is clear that long-exon variants are showing more NMD-efficiency compared to start-proximal and distal sites. So even though the authors discussed about long-exon regions in the discussion, they should expand it with 1-2 sentences with more NMD efficiency seen variants in those regions.

2) Also, in Figure 1d, although the legend has been corrected, the y-axis label still includes VUS. So I think they should remove it.

(Remarks on code availability)

The code seems more organized and the README file is clear.

made.

Manuscript title: "Regional nonsense constraint offers biological and clinical insights into genetic disease"

We would like to thank all the reviewers for their considerate feedback and for recognising the importance of our work. We offer a point-by-point response to each of their comments below.

Reviewer 1

Blakes et al have developed a regional nonsense constraint methodology that confirms some of our fundamental understandings of NMD after PTCs are introduced. Importantly, this new measure can be used to improve the interpretation of LoF variants in rare disease diagnostics. In addition, the authors highlight a major limitation in current measures of pLoF constraint such as LOEUF that are used extensively by the genetics community to interpret potential diagnostic variants in rare disease cases and to drive discovery of new disease-gene associations. These current measures treat the whole transcript as having an equal chance of harbouring a LoF variant following a PTC event but 40% of the coding genome is predicted to escape NMD. Their new metric is capable of more accurately identifying genes under constraint and they highlight 25 new candidate disease genes with compelling evidence from their new metric and clusters of *de novo* variants in rare disease cases from the 100,000 Genomes Project. This is an important study that will of interest to the whole rare disease community and I just have a few comments/suggestions.

1. They describe existing guidelines (ref 21) to accurately interpret PTCs using the ACMG guidelines and these already take into account NMD-target or escape regions. Some more discussion of how using their metric is better for diagnostic interpretation than just following these guidelines would be welcome or, better still, how their metric could be used to improve the ACMG guidelines.

We thank the reviewer for this suggestion. We have expanded the discussion to address this point (**Lines 305-310**).

2. There are likely to be existing false positive diagnoses where, historically, clinical geneticists may not have considered whether NMD escape was likely and they would not have had these new measures of regional constraint. For example, the standard 100,000 Genomes Project pipeline highlights tier 1 LoF variants in known disease genes related to a patient's conditions and many of these resulted in a diagnosis. Could they identify potential examples in the 100,000 Genomes Project data or give estimates or discuss the likelihood of their approach "fixing" false positive diagnoses and implications.

We agree there may be opportunities to address false-positive diagnoses in principle. For example, in **Extended Data Figure 10** we show that almost 20% of P/LP nonsense and frameshift variants in ClinVar fall in unconstrained regions. We appreciate the suggestion of exploring potential false positive diagnoses in the 100KGP. However, to validate or overturn clinical diagnoses would require the recall of patients and extensive functional and clinical work which are not currently feasible. The data we present in **Figure 6a** argue against systematic over-diagnosis for PTCs in unconstrained regions in 100KGP. For example, the odds of receiving a diagnosis are the same for individuals with a *de novo* PTC in an unconstrained region or a *de novo* synonymous variant. The examples of *GATA6* and *KANSL1* (**Figure 5**) also highlight known dominant disease genes where variants in specific regions should be interpreted with caution.

3. Some of the results around long-exon variants (Figure 1d) seemed to suggest they were more likely to be P/LP and were highly constrained and behaved more like NMD-target rather than NMD-escape variants? It was a bit unclear to me how to interpret such variants in the future and this could be clarified more in the discussion.

We agree that these results can be counterintuitive. Our interpretation of these data is that variants in long-exon regions, even if they do escape NMD, will invariably truncate a very large proportion of the CDS. Therefore, mechanistically, they are likely to cause loss of function. We have added further discussion on this point (**Lines 285-293**).

4. Related to this, much of the analysis relies, understandably, on ClinVar classification of LoF variants into P/LP or VUS but how reliable is this given that probably many would not have been validated for being true LoF variants?

We agree that ClinVar has important limitations. Nevertheless, it is widely used as a variant classification benchmark in our field. To avoid over-reliance on ClinVar we diversified our analyses to include clinical-grade diagnostic outcome data (**Figure 6a**) and variant enrichment statistics which are agnostic to clinical curations (**Figure 6b-e**).

Reviewer 2

Blakes and colleagues describes a new metric for interpreting genetic variants based on nonsense mediated mRNA decay (NMD) based on comprehensive population sequencing data. Using this metric, the authors identify ~3k genes that have regional constraint based on the prevalence of mutations, which they use to interpret trio data, ClinVar, etc. The authors did a nice job of code/data sharing for their annotations. Overall, this seems to be a useful metric and is nicely executed. I have one major critique that should be addressed and a few minor points, but would suggest this work seems close to ready for publication.

5. The glaring hole in this study, in my view, is the lack of sufficient transcriptomics data that would verify the NMD mechanism as the putative driver. My understanding is that GEL has RNA for most individuals. Otherwise, 1000G would have paired RNA and DNA for every donor. My suggestion is that the authors scan the genomes from these data sources and identify mutations predicted to trigger NMD. Then, looking in the RNA-seq data, one could perform an allele-specific abundance analysis in RNA. If the metrics that they develop are correct, one should see a significant depletion of the NMD-triggering RNA.

We thank the reviewer for this excellent suggestion. We have now performed a transcriptome-wide allele-specific expression analysis for nonsense variants across NMD regions using RNA-Seq data in blood from 5,132 rare disease probands in the 100KGP. Broadly, these data show stronger allele-specific expression for variants in NMD-target regions than our proposed NMD-escape regions, consistent with increased NMD efficiency in NMD-target regions. We present these data in a new figure (**Figure 2**).

6. Lines 130-134 should be accompanied with statistical analyses.

We have added denominators and percentages for the number of newly enriched genes compared to GeneBayes (**lines 142-143**). These statistics are purely descriptive and are not intended to be framed in terms of a testable hypothesis, so we have omitted statistical comparisons at present.

7. I don't understand the heterozygous only / non-homozygous limitation (line ~290). Could the authors clarify why this is the case and comment on the occurrence of homozygous-driven diseases (as they note in the introduction with heterozygous variant causes of disease).

In this sentence we wanted to convey that for variants that are only pathogenic in the biallelic state (i.e. pathogenic for recessive conditions), the strength of selection against heterozygous variants is expected to be weak. We would therefore have very limited power to detect

constraint against these variants. This is a limitation of most population constraint metrics. We have now deleted this sentence to avoid confusion.

8. The code looks roughly complete but has a glaring lack of any sort of documentation to help an outside reader. The code resource should be revised before publication.

We have updated the codebase and the documentation in the GitHub repository (https://github.com/alexblakes/regional_nonsense_constraint).

Reviewer 3

This study by Blakes et al. investigates how the position of premature termination codons (PTCs) within genes influences their degradation by nonsense-mediated decay (NMD), a key cellular quality control mechanism. The authors show that ~one-third of the human coding genome is predicted to escape NMD, potentially producing truncated proteins that may cause disease via dominant-negative or gain-of-function effects. Using population-scale exome data (gnomAD v4.1) and a novel regional constraint metric (OE95), they identify 2,764 transcripts with significant intolerance to PTCs in specific transcript regions—some of which are missed by existing metrics like LOEUF. These constrained regions are enriched for de novo truncating variants in rare disease trios, with stronger enrichment correlating with higher diagnostic yield. The study highlights 25 genes as novel disease candidates based on clustered de novo PTCs in constrained regions. Overall, the findings refine the interpretation of pLoF variants, improve variant pathogenicity classification, and offer new insights into disease gene discovery. Although the study provides important results in PTC-variant interpretation field, I have some major concerns.

9. Although the canonical rule is the most established rule for variability in NMD efficiency, the other two rules (start-proximal < 150 bp) and the long exon rules are not robustly validated especially for the germline variants. Lindeboom et al. (2016, 2019) papers put forward those two non-canonical rules mainly for somatic variants. Even for somatic variants, the NMD efficiency varies a lot within those regions, for instance for start-proximal PTCs that have in-frame downstream translation initiation site are more likely to be NMD escapees due to alternative translation initiation mechanism. For the long-exon rule, NMD efficiency greatly varies based on the distance from PTC to the end of the transcript. So for these two rules, it is not optimal to consider every PTC in those regions are NMD-escapees especially for germline variants. The authors should deep dive into these rules carefully, at least for additional signals such as in-frame downstream translation initiation site or PTC distance to the exon end. In this way, they can be more informative.

We thank the reviewer for their insightful comment. Several features are known to modify NMD efficiency (e.g. upstream open reading frames, non-canonical start codons, Kozak consensus sequence strength, 3' UTR length, etc). The focus of our article is on utilising the positional NMD rules because they are the strongest determinants of NMD efficiency and because they optimise for interpretability in a clinical context. We acknowledge the need to validate these rules in a germline context. We have therefore performed an extensive analysis of allele-specific expression for nonsense variants across NMD regions in RNA-Seq data in blood (**see response #5 to Reviewer #2**).

10. Figure 1D – Are these the normalized counts retrieved from all of the ClinVar nonsense/frameshifts or just the pathogenic(P)/likely pathogenic (LP) ones? Could the authors clarify about that?

In **Figure 1d** we show all nonsense / frameshift variants in ClinVar, not just P/LP variants. We have updated the figure legend to clarify this point.

11. Although the start-proximal, long-exon and distal regions are only forming a smaller part of the gene/transcript (Figure 1c), how the genes are constrained in those regions are the genes

mostly constrained overall according to pLI or LOEUF scores? It seems that the genes that they identified that are constrained in any region (~2.7K) are mostly identified as constrained by pLI or LOEUF. What could be expected that there are a substantial number of genes that are constrained in those small regions, but not constrained at all. Could the authors give an explanation to that?

We thank the reviewer for highlighting this point. We assume that the reviewer is referring to the first Venn diagram in **Figure 3c** where we show that 2,446/2,764 (11.5%) genes constrained in 'any region' have a high pLI/LOEUF score. This is expected because 'any region' encompasses, start-proximal, long-exon and distal regions, as well as NMD-target regions, which comprise the largest part of the genomic footprint of most genes.

12. Although this metric is only developed for nonsense variants, the authors also analysed the enrichment of frameshift variants in rare disease trios (Figure 5). I think frameshift variants greatly differ in terms of their NMD escape region definition. So the analysis of frameshift variant using a metric developed for nonsense variants is misleading.

We thank the reviewer for raising this important point. EJC-mediated NMD is determined largely by the position of the PTC in the transcript. Whether that PTC is introduced by a nonsense variant, a splicing variant, or a frameshift variant is secondary. Although EJC-mediated NMD is a co-translational process, it is dependent on the structure of the transcript and not the amino acid sequence of the translated protein. Frameshift variants do not, therefore, differ in their NMD-escape region definitions *per se*. Where they do differ is that they introduce a PTC downstream of the position of the indel. We have therefore updated our analysis of *de novo* frameshift variants with their NMD-region determined by the position of the downstream PTC (see **Annotation of de novo frameshift variants with Aenmd** in **Methods**). We find that 17% of *de novo* frameshift variants in our trio cohorts produce a PTC in a different NMD-region to that of the indel. We have updated our analyses of these variants accordingly (**Figure 6**).

13. How about the normalization procedure by the gene length? The authors did do it by the normalization of synonymous variants but they should try it by the length of region as well and compare it.

The constraint model accounts for both mutation rate and sequence length (see **Regional nonsense constraint** in **Methods**). We effectively sum the Roulette mutation rate for every possible SNV per transcript / region / variant class. We have updated the text of the methods to clarify this further (**Lines 427-428**).

14. I reviewed the website for the code but I think some parts in the code are missing or outdated. For instance, could not find how they calculated the observed/expected ratio for each gene/transcript.

We have updated the code base and the accompanying documentation in our GitHub repository (https://github.com/alexblakes/regional_nonsense_constraint).

15. The feature analysis of long-exon, distal sites and start-proximal sites could be more deeper including the protein structural domain analysis and some other potential protein features.

We thank the reviewer for this suggestion. We have updated our analyses to explore the distribution of Pfam domains in NMD regions transcriptome wide (**lines 154-160**). These analyses are presented in **Extended Data 8** and show that constrained NMD-target and distal regions are more likely to overlap Pfam domains than unconstrained regions. We are under-powered to detect these differences in start-proximal or long exon regions.

16. It is also not clear that the authors just focused on the transcripts with at least 150 bp and/or long exon. For most of the short transcripts, these will not be available. And for some of transcripts (≤ 2 exons), most of transcript will be NMD-escape. How did they evaluate those short transcripts? Did they exclude them?

We included all protein-coding genes from GENCODE in our analysis. We did not exclude any genes based on transcript length or structure. Where a gene lacked an NMD region (e.g. single exon genes lacking an NMD-target region, or genes without a long exon $>400\text{nt}$), we did not provide a constraint annotation for that region. We have updated the **Methods** to clarify these points (**lines 334-335**).

17. The code is a usable resource for the community but some parts in the code are missing or outdated. For instance, could not find how they calculated the observed/expected ratio for each gene/transcript.

As above, we have now updated the code base and the accompanying documentation in our shared GitHub repository (https://github.com/alexblakes/regional_nonsense_constraint).

Manuscript title: "Regional nonsense constraint offers biological and clinical insights into genetic disease"

We thank all the reviewers for their comments.

Reviewer 1

I am satisfied that the authors have addressed my questions and added relevant new detail where required. They have also significantly improved the study based on the feedback from the other reviewers by incorporating transcriptomic analysis and fixing the code availability issues.

Reviewer 2

The revisions have addressed all of my questions. I have no major concerns remaining.

Reviewer 3

The reviewers addressed my questions in a very detailed manner.

I think their results are overall noteworthy to be published and will be of high significance to the variant interpretation field. But I have two more concerns to be addressed as you can see below:

- 1) Although the start-proximal rule and long-exon rules have been shown before and now by the authors (Figure 2) associated with less NMD-efficiency, we see some variability among variants in those regions. So when we call all those regions as NMD-escape, that can be misleading especially for long-exon regions. Also as you can see the results from the Figure 2, it is clear that long-exon variants are showing more NMD-efficiency compared to start-proximal and distal sites. So even though the authors discussed about long exon regions in the discussion, they should expand it with 1-2 sentences with more NMD efficiency seen variants in those regions.

Thank you for raising this point. We have added a sentence to the discussion to emphasise that NMD efficiency is generally weaker in long exon regions than other putative NMD-escape regions.

- 2) Also, in Figure 1d, although the legend has been corrected, the y-axis label still includes VUS. So I think they should remove it.

The figure displays the proportion of all variants which are VUS. We have updated the axis title for clarity.